# Vaccination coverage and adverse events following a reactive vaccination campaign against hepatitis E in Bentiu displaced persons camp, South Sudan

Robin C. Nesbitt[1]*, Vincent Kinya Asilaza[2], Etienne Gignoux[1,3], Aybüke Koyuncu[1,4], Priscillah Gitahi[2], Patrick Nkemenang[2], Jetske Duncker[2], Zelie Antier[2], Melat Haile[3], Primitive Gakima[3], Joseph F. Wamala[5], Fredrick Beden Loro[5], Duol Biem[6], Monica Rull[3], Andrew S. Azman[3,4,7,8], John Rumunu[6], Iza Ciglenecki[3]

1 Epicentre, Paris, France, 2 Médecins Sans Frontières, Juba, South Sudan, 3 Médecins Sans Frontières, Geneva, Switzerland, 4 Johns Hopkins University Bloomberg School of Public Health, Baltimore, United States of America, 5 World Health Organization, Juba, South Sudan, 6 Ministry of Health, Juba, South Sudan, 7 Geneva Centre for Emerging Viral Diseases, Geneva University Hospitals, Geneva, Switzerland, 8 Division of Tropical and Humanitarian Medicine, Geneva University Hospitals, Geneva, Switzerland

* robin.nesbitt@epicentre.msf.org

**Data Availability Statement:** The minimal data set underlying the findings of this paper are available

## Abstract

### Introduction

Hepatitis E (HEV) genotypes 1 and 2 are the common cause of jaundice and acute viral hepatitis that can cause large-scale outbreaks. HEV infection is associated with adverse fetal outcomes and case fatality risks up to 31% among pregnant women. An efficacious three-dose recombinant vaccine (Hecolin) has been licensed in China since 2011 but until 2022, had not been used for outbreak response despite a 2015 WHO recommendation. The first ever mass vaccination campaign against hepatitis E in response to an outbreak was implemented in 2022 in Bentiu internally displaced persons camp in South Sudan targeting 27,000 residents 16–40 years old, including pregnant women.

### Methods

We conducted a vaccination coverage survey using simple random sampling from a sampling frame of all camp shelters following the third round of vaccination. For survey participants vaccinated in the third round in October, we asked about the onset of symptoms experienced within 72 hours of vaccination. During each of the three vaccination rounds, passive surveillance of adverse events following immunisation (AEFI) was put in place at vaccination sites and health facilities in Bentiu IDP camp.

### Results

We surveyed 1,599 individuals and found that self-reported coverage with one or more dose was 86% (95% CI 84–88%), 73% (95% CI 70–75%) with two or more doses and 58% (95% CI 55–61%) with three doses. Vaccination coverage did not differ significantly by sex or age

on request, in accordance with the legal framework set forth by Médecins Sans Frontières (MSF) data sharing policy (Karunakara U, PLoS Med 2013). MSF is committed to sharing and disseminating health data from its programs and research in an open, timely, and transparent manner to promote health for populations while respecting ethical and legal obligations towards patients, research participants, and their communities. The MSF data sharing policy ensures that data will be available upon request to interested researchers while addressing all security, legal, and ethical concerns. All readers may contact the MSF generic address data.sharing@msf.org, or the Epicentre generic address epimail@epicentre.msf.org to request the data that can be shared with researchers subject to the establishment of a data sharing agreement to provide the legal framework for data sharing.

**Funding:** The author(s) received no specific funding for this work.

**Competing interests:** The authors have no competing interests to disclose.

group. We found no significant difference in coverage of at least one dose between pregnant and non-pregnant women, although coverage of at least two and three doses was 8 and 14 percentage points lower in pregnant women. The most common reasons for non-vaccination were temporary absence or unavailability, reported by 60% of unvaccinated people. Passive AEFI surveillance captured few mild AEFI, and through the survey we found that 91 (7.6%) of the 1,195 individuals reporting to have been vaccinated in October 2022 reported new symptoms starting within 72 hours after vaccination, most commonly fever, headache or fatigue.

## Conclusions

We found a high coverage of at least one dose of the Hecolin vaccine following three rounds of vaccination, and no severe AEFI. The vaccine was well accepted and well tolerated in the Bentiu IDP camp community and should be considered for use in future outbreak response.

### Author summary

Hepatitis E virus can cause large, protracted outbreaks in populations with limited access to safe water and sanitation. Hepatitis E infection is particularly dangerous for pregnant women in their third trimester. A vaccine, Hecolin, exists but it was never used in outbreak response despite a recommendation from the World Health Organization. In 2022, the Ministry of Health of South Sudan and Médecins Sans Frontières used the vaccine during an outbreak for the first time in an internally displaced persons camp with more than 100,000 people. We report on the results of the vaccination campaign, which show a high uptake of the vaccine and no serious side effects. We show that a vaccination campaign with Hecolin during an outbreak can reach many people at risk and is well tolerated. This will hopefully inspire confidence to use this vaccine again in the future.

## Introduction

Hepatitis E virus (HEV) genotypes 1 and 2 (g1/g2) have been reported throughout the world, though outbreaks have mostly been reported in populations with precarious access to water and sanitation, such as refugee and displaced persons camps. HEV g1/g2 infection causes jaundice and is usually self-limiting, although in a small proportion of cases, infection can cause acute fulminant hepatitis [1]. HEV g1/g2 infection is associated with negative pregnancy outcomes such as stillbirth [2] and maternal clinical case fatality ratios of up to 31% among pregnant women [3–6], particularly in the third trimester. It was estimated that there were 3.4 million symptomatic cases in 2005 [7], however limited access to diagnostics and lack of surveillance means that the burden of morbidity and mortality is likely underestimated.

Hecolin (HEV 239, Innovax, China) is a recombinant vaccine based on HEV genotype 1 and has been licensed in China since 2011 for adults over 16 years. In a phase three trial with more than 110,000 participants, the efficacy of Hecolin was 100% (95% CI 72.2–100) for the full three-dose schedule over the first 19 months [8] and 86.8% (95% 71–94%) for at least one dose over the first 4.5 years after first vaccination [9]. Although not recommended for use in routine immunization programmes, the WHO issued a statement in 2015 and again in 2021 recommending the use of hepatitis E vaccine (HEV 239, Hecolin) to mitigate or prevent

outbreaks, including the vaccination of pregnant women [10,11]. Until 2022, the vaccine had not been used in response to an outbreak.

In March 2022, the Ministry of Health (MoH) in South Sudan and Médecins Sans Frontières (MSF) implemented the first mass reactive vaccination campaign against hepatitis in Bentiu internally displaced persons (IDP) camp, South Sudan [12]. Bentiu IDP camp was established in 2014 as a Protection of Civilians site at a United Nations Mission in South Sudan base in response to conflict in Unity State. Cases of hepatitis E were detected in the camp since it was established, with large outbreaks occurring in 2015 (2,189 cases reported) and in 2016 (924 cases reported). HEV transmission continued despite increasingly formalised infrastructure and coordinated humanitarian response, and an outbreak was declared by the MoH in August 2021. In response, the MoH and MSF conducted a vaccination campaign in three rounds in March, April and October 2022. MSF and the MoH put in place a series of operational research studies aimed to document the feasibility, and safety of the vaccination.

The vaccination campaign targeted 26,848 residents of Bentiu IDP camp aged 16–40 years old, including pregnant women. This age group was chosen to benefit from the limited vaccine doses available and because the vaccine is not registered for use in children younger than 16 years; the upper limit was chosen based on pre-vaccination incidence rate estimates, which were lower among older adults. Individuals with jaundice, known chronic liver disease, immunodeficiency or other acute severe illness were excluded from vaccination. During the second and third rounds, the vaccine was offered to anyone within the target group (16–40 years old and residence in Bentiu IDP camp), regardless of whether they received a previous dose or not. The campaign used a combined strategy of fixed vaccination points and mobile vaccination teams at markets, food distribution points and water points [12]. The administrative coverage (i.e., number of doses delivered divided by the target population size, 26,848 individuals) was over 90% for all three vaccination rounds: the first round in March reached 24,469 people (91% of the target population), the second round in April reached 25,434 people (95% of the target population), the last round in October reached 30,264 people (113% of the target population, Table 1).

Here we describe the results of a survey aimed to estimate vaccination coverage within the targeted population immediately after the third and final vaccination round. We additionally aimed to document adverse events following immunization (AEFI) via passive surveillance and estimate the proportion of people who reported to have experienced new symptoms after vaccination in the survey.

## Methods

### Ethics

Ethical approval was granted for this survey from the Médecins Sans Frontières Ethical Review Board and by the South Sudan Ministry of Health Research Ethics Review Board as part of the study protocol titled: "Effectiveness, safety and feasibility of recombinant hepatitis E vaccine HEV 239 (Hecolin) during an outbreak of hepatitis E in Bentiu, South Sudan." Approval numbers are MSF ERB #2167 and RERB-MOH # 54/27/09/2022.

All participants provided verbal informed consent before participating in the interview.

### Representative vaccination coverage survey

Following the third round of vaccination, we conducted a coverage survey with a target sample size of 836 vaccine-eligible individuals. We calculated the sample size using Emergency Nutrition Assessment software (ENA for SMART, 2020 version) with the following parameters: 5% precision around the estimated three-dose vaccine coverage estimate of 50%, considering a

**Table 1. Vaccination campaign dates, target population, results and administrative coverage by round and by dose, Bentiu IDP camp, South Sudan, 2022.**

|  | Dates | Target population | Vaccinated | 1st dose | 2nd dose | 3rd dose | Administrative coverage |
|---|---|---|---|---|---|---|---|
| Round 1 | 22–30 March | 26,848 | 24,469 | 24,469 |  |  | 91% |
| Round 2 | 19–26 April | 26,848 | 25,434 | 5,573 | 19,861 |  | 95% |
| Round 3 | 4–25 October | 26,848 | 30,264 | 9,722 | 6,249 | 14,293 | 113% |
| Total n |  | 26,848 | 80,167 | 39,764 | 26,110 | 14,293 |  |
| Coverage % |  |  |  | 148% | 97% | 53% |  |

conservative design effect (DEFF) of 2 due to within household correlation in coverage. We estimated that 590 households were required to achieve this sample size based on an average household size of 7 people, 2 vaccine-eligible individuals 16–40 years per household, and a 10% refusal ratio. The target population for the survey was vaccine-eligible individuals: 16–40 years old at the time of the survey conducted immediately after the third and final vaccination round, resident of Bentiu IDP camp (excluding Bentiu town and other communities outside the camp), and without jaundice, known chronic liver disease, immunodeficiency or any acute illness during the vaccination campaign. All vaccine-eligible individuals in each household were eligible for inclusion in the survey.

In total, 600 shelters were selected by simple random sampling without replacement using the R sampling package 'srswor' from a sampling frame of all 12,139 shelters in Bentiu IDP camp, provided by International Organization for Migration (IOM) camp management. All households living in the selected shelter were eligible for inclusion.

A team of seven data collectors and one supervisor were trained and interviews took place for a total of 11 days (28 October until 9 November, excluding Sundays). For each household, after obtaining verbal consent from the head of household, we asked about all members of the household, including age and sex, to determine household size and identify vaccine-eligible individuals. For all members of the household, we asked whether they had experienced jaundice or been diagnosed with hepatitis E since 2014, when Bentiu IDP camp was established. For household members 16–40 years old at the time of survey, vaccination status was determined through interviews and checking vaccination cards. Responses were accepted from the head of household or a delegated person over 18 years old if vaccine-eligible individuals were not present at time of interview. Photographs were taken of vaccination cards, when available and consented. The team conducted up to two revisits if household residents were absent on the day of the interview.

Questionnaires were conducted using Open Data Kit (ODK) Collect (https://getodk.org/) on mobile devices. Data were uploaded to a secure server every evening; quality checks were done and feedback was given to the data collection team daily.

## Pharmacovigilance / Adverse Events Following Immunisation (AEFI)

Prior to the first round of vaccination, passive surveillance of adverse events following immunisation (AEFI) was put in place at all 5 primary health care centers (PHCCs) inside Bentiu IDP camp, the MSF hospital and at the vaccination sites. Paper registers and forms were distributed, and facility staff were trained to complete the register and the reporting forms when a patient reported symptoms following vaccination. During and after each vaccination round, MSF staff visited the PHCCs to collect the completed registers and forms. During the first round, vaccinees were asked to stay at the sites for 15 minutes observation period after vaccination. As the vaccination campaign became more mobile in the subsequent rounds, individuals were not required to wait at the site, they were instead encouraged to go to the clinic for

any symptoms. Passive AEFI surveillance systems likely do not capture mild illness; we therefore asked survey respondents reporting to have been vaccinated in the third vaccination round whether they experienced the onset of new symptoms within 72 hours of vaccination.

## Statistical analysis

All indicators (i.e., sex, age, and demographic characteristics) were calculated as proportions, and differences in proportions were evaluated using Pearson χ2 tests. Vaccination coverage, 95% confidence intervals (95% CI), considering household clustering, and household design effects were estimated using the survey package in R.

# Results

## Survey population

From the 600 shelters randomly selected for interview, 5 (0.8%) were abandoned (unoccupied for a long time), 34 (5.7%) were absent after two revisits and 1 (0.17%) household refused to participate. In total, we conducted interviews at 560 shelters (95% of the target) including 4,057 individual residents. Among them, 1,669 (200% of the target sample size) vaccine-eligible individuals were identified. The mean shelter size was 7.2, with a reported average of 4.2 vaccine-eligible individuals aged 16–40 years per shelter. Out of 1,669 vaccine-eligible individuals, 1,599 (95%) reported their vaccination status.

The person responding to the interview was the head of household or a delegated adult for 83.5% of included individuals overall (this includes delegated reporting for children, counted as members of each household), and 68% of vaccine-eligible individuals (Table 2). This differed by sex as women were more likely to be found at home, with 43% of women responding directly compared to 17% of men.

## Previous hepatitis E infection

Overall few individuals reported previous jaundice or hepatitis E infection (n = 64, 1.6%) since 2014 (when Bentiu IDP camp was established). Among those, 89% reported seeking care anywhere and 75% (n = 48) sought care at the MSF hospital.

## Coverage results

We estimate that 58% (95% CI 55–61, Design effect [DEFF] = 1.7) of the vaccine-eligible population had three doses of Hecolin, 15% (95% CI 13.0–16.7%, DEFF = 1.2) had two doses and 14% (95% CI 11.7–15.6%, DEFF = 1.3, Table A in S1 text) had one dose. This translates into 86% (95% CI 84–88, DEFF = 1.6) of the vaccine-eligible population having at least one dose, and 73% (95% CI 70–75%, DEFF = 1.6) having at least two doses (Table 3). Among those reporting to have been vaccinated, less than half were able to provide a vaccination card (Table 3 and Table A in S1 text). Calculated design effects show that vaccination status was correlated within households.

We found no significant differences in vaccination coverage by sex, age-group or by location of residence within the camp (Table 4, Fig 1). Among 118 pregnant women 117 had a known vaccination status; vaccination coverage with at least 2 doses and complete 3 dose vaccination coverage were significantly lower than among non-pregnant women (at least 2 doses: 66% (95% CI 57–74) vs non-pregnant women 74% (95% CI 71–78), p = 0.05, 3 doses: 44% (95% CI 34.0–53.2) vs. non-pregnant women 58% (95% CI 53.7–61.5), p = 0.006, Table 4).

We conducted a sensitivity analysis including only individuals who reported their vaccination status directly for themselves and found similar overall vaccination coverage for partial

**Table 2. Survey sample population characteristics by sex, n = 4,057 individuals (n = 1,599 known vaccination status).**

| | Female | Male | Overall |
|---|---|---|---|
| | n (%) | n (%) | n (%) |
| **Age group (years)** | | | |
| 0-5 | 312 (14.7%) | 338 (17.5%) | 650 (16.0%) |
| 6-10 | 315 (14.8%) | 336 (17.4%) | 651 (16.1%) |
| 11-15 | 329 (15.5%) | 382 (19.8%) | 711 (17.5%) |
| 16-24 | 464 (21.8%) | 425 (22.0%) | 889 (21.9%) |
| 25-40 | 485 (22.8%) | 295 (15.3%) | 780 (19.2%) |
| 41+ | 220 (10.4%) | 156 (8.1%) | 376 (9.3%) |
| Total | 2125 (100%) | 1932 (100%) | 4057 (100%) |
| **Pregnant[1]** | | | |
| No | 1049 (89.8%) | - | 1049 (89.8%) |
| Yes | 119 (10.2%) | - | 119 (10.2%) |
| **Trimester of pregnancy** | | | |
| 1 | 23 (19.3%) | - | 23 (19.3%) |
| 2 | 50 (42.0%) | - | 50 (42.0%) |
| 3 | 46 (3.7%) | - | 46 (3.7%) |
| **Education[2]** | | | |
| None | 808 (38.0%) | 265 (13.7%) | 1073 (26.5%) |
| Primary | 300 (14.1%) | 355 (18.4%) | 655 (16.1%) |
| Secondary | 60 (2.8%) | 226 (11.7%) | 286 (7.1%) |
| University | 1 (0.05%) | 30 (1.6%) | 31 (0.8%) |
| NA | 956 (45.0%) | 1056 (54.7%) | 2012 (49.6%) |
| **Employment** | | | |
| Cook | 9 (1.0%) | 2 (0.3%) | 11 (0.7%) |
| Driver | 0 (0) | 1 (0.2%) | 1 (0.06%) |
| Logistics | 0 (0) | 6 (0.9%) | 6 (0.4%) |
| Health or medical | 1 (0.1%) | 9 (1.4%) | 10 (10.6%) |
| Student | 43 (4.6%) | 75 (11.4%) | 118 (7.4%) |
| Unemployed | 878 (93.3%) | 546 (83.0%) | 1424 (89.1%) |
| Unknown | 1 (0.1%) | 1 (0.2%) | 2 (0.1%) |
| **Survey respondent** | | | |
| Delegate | 1615 (76.0%) | 1774 (92.0%) | 3383 (83.0%) |
| Self | 508 (23.9%) | 158 (8.2%) | 666 (16.4%) |
| Unknown | 2 (0.1%) | 0 (0) | 2 (0.05%) |
| **Hepatitis E Vaccination card[3]** | | | |
| None | 484 (51.4%) | 471 (71.6%) | 955 (59.7%) |
| >= 1 | 457 (48.6%) | 187 (28.4%) | 644 (40.3%) |

[1] women of childbearing age 14–45 years old

[2] pregnant women

[3] adults > = 16 years

[4] individuals 16–40 years old with known vaccination status

and complete vaccination (Table B in S1 text). Similarly in a second sensitivity analysis excluding 40-year-olds to account for possible misreporting of age, we also found similar vaccination coverage (Table C in S1 text).

**Table 3. Vaccination coverage by dose according to recall and confirmed by card, n = 1,599.**

|  | According to recall or card |  |  | Confirmed by card |  |  |
|---|---|---|---|---|---|---|
| Coverage by dose | % (n) | 95% CI | DEFF | % (n) | 95% CI | DEFF |
| One or more doses | 86% (1377) | [84–88] | 1.6 | 40% (644) | [37–43] | 1.8 |
| Two or more doses | 73% (1160) | [70–75] | 1.6 | 19% (305) | [17–21] | 1.3 |
| Three doses | 58% (924) | [55–61] | 1.7 | 10% (163) | [9–12] | 1.2 |

Note that confirmed by card means that all doses reported were verified on vaccination card and those without a card were considered unvaccinated. DEFF = design effect.

According to the population distribution in selected households participating in the survey, the vaccine-eligible population accounted for 41% of the total camp population (Table 2, S1A Fig). The high coverage among the eligible population therefore resulted in only 35% partial coverage with at least one dose, 30% with at least two doses and 24% with the full three doses in the full camp population.

## Reasons for non-vaccination

The most common reasons for non-vaccination were absence or unavailability, given by 60% of unvaccinated people, 80% of those who missed their second dose and 77% of those who missed their third dose (Table 5). Not living in Bentiu IDP camp at the time of previous

**Table 4. Vaccination coverage according to recall stratified by sex, pregnancy status, age group and sector, n = 1,599.**

|  | One or more doses |  |  | Two or more doses |  |  | Three doses |  |  |
|---|---|---|---|---|---|---|---|---|---|
|  | % (n) | 95% CI | DEFF | % (n) | 95% CI | DEFF | % (n) | 95% CI | DEFF |
| **Sex** |  |  |  |  |  |  |  |  |  |
| Male | 85% (557) | [81–88] | 1.6 | 72% (471) | [67–76] | 1.7 | 61% (399) | [56–65] | 1.7 |
| Female | 87% (820) | [85–90] | 1.3 | 73% (689) | [70–76] | 1.4 | 56% (525) | [52–60] | 1.4 |
| **p-value** | **0.20** |  |  | **0.55** |  |  | **0.10** |  |  |
| **Pregnancy** |  |  |  |  |  |  |  |  |  |
| Pregnant | 84% (98) | [76–90] | 1.0 | 66% (77) | [57–74] | 1.0 | 44% (51) | [34–53] | 1.1 |
| Not pregnant | 88% (721) | [85–90] | 1.2 | 74% (611) | [71–78] | 1.3 | 58% (474) | [54–61] | 1.3 |
| **p-value** | **0.23** |  |  | **0.05** |  |  | **0.006** |  |  |
| **Age group** |  |  |  |  |  |  |  |  |  |
| [16,20] | 88% (541) | [85–91] | 1.3 | 72% (440) | [67–76] | 1.4 | 57% (350) | [52–62] | 1.4 |
| (20,25] | 83% (264) | [78–88] | 1.3 | 70% (221) | [64–75] | 1.3 | 56% (177) | [49–62] | 1.4 |
| (25,30] | 84% (193) | [79–89] | 1.2 | 70% (162) | [64–77] | 1.1 | 56% (128) | [49–63] | 1.2 |
| (30,35] | 85% (148) | [79–90] | 1.1 | 75% (132) | [69–82] | 1.1 | 63% (110) | [56–70] | 1.0 |
| (35,40] | 89% (231) | [85–93] | 1.1 | 79% (205) | [74–84] | 1.0 | 61% (159) | [55–67] | 1.0 |
| **p-value** | **0.43** |  |  | **0.18** |  |  | **0.17** |  |  |
| **Sector** |  |  |  |  |  |  |  |  |  |
| Sector 1 | 83% (193) | [74–90] | 2.6 | 70% (162) | [60–78] | 2.2 | 55% (127) | [46–64] | 2.0 |
| Sector 2 | 83% (233) | [76–88] | 1.6 | 66% (185) | [59–72] | 1.4 | 52% (147) | [45–59] | 1.5 |
| Sector 3 | 90% (357) | [86–93] | 1.2 | 76% (301) | [71–80] | 1.3 | 63% (251) | [57–69] | 1.5 |
| Sector 4 | 85% (215) | [78–90] | 1.7 | 70% (179) | [62–78] | 2.1 | 53% (134) | [44–61] | 2.0 |
| Sector 5 | 87% (379) | [83–90] | 1.3 | 77% (333) | [72–81] | 1.3 | 61% (265) | [55–67] | 1.7 |
| **p-value** | **0.19** |  |  | **0.078** |  |  | **0.088** |  |  |

DEFF = design effect.

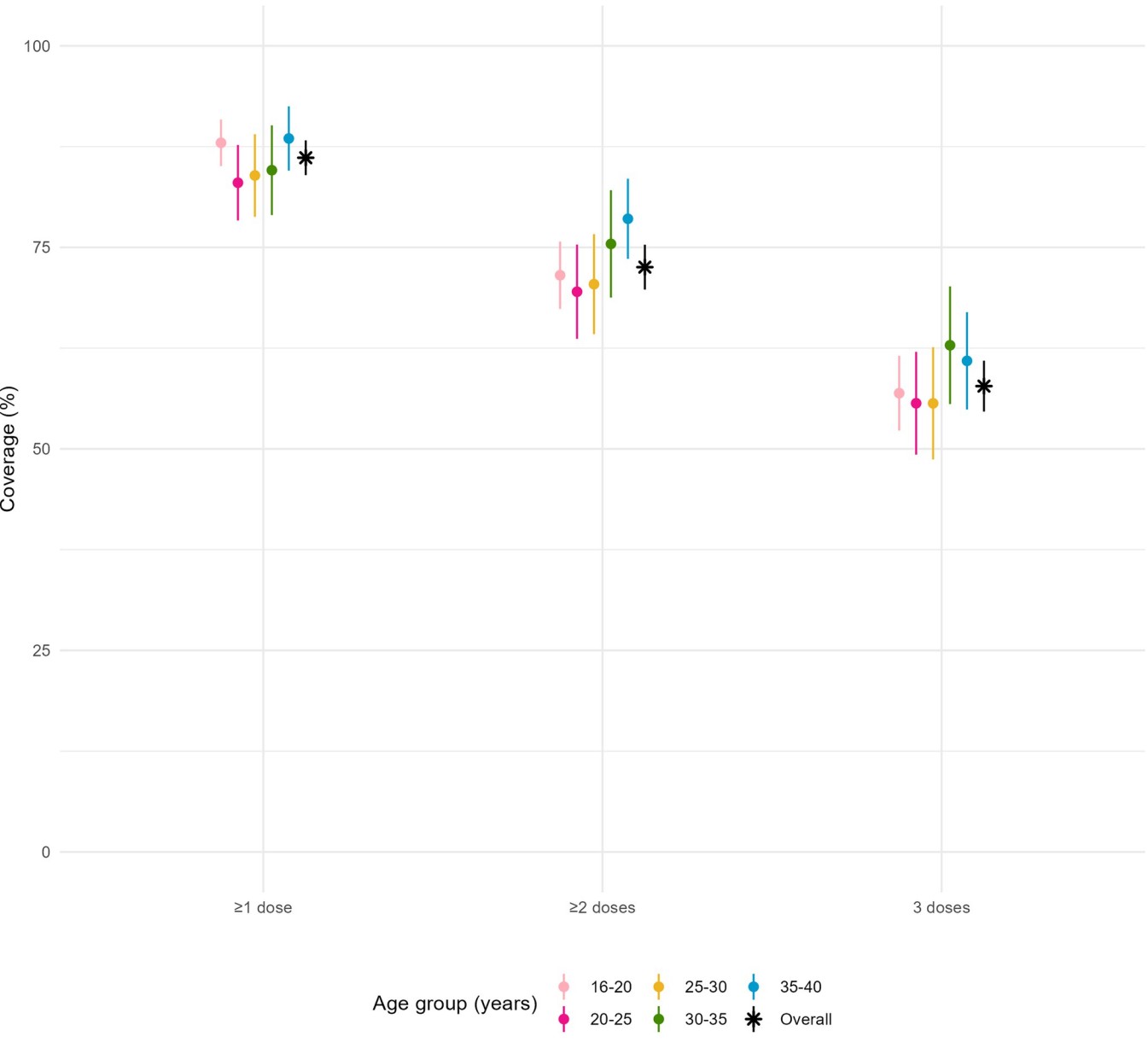

**Fig 1. Vaccination coverage according to recall by dose and age group.**

vaccination rounds was reported by 20.6% of non-vaccinated individuals, 27.5% of those with one dose and 11.6% of those with two doses. The proportion of individuals not vaccinated due to illness at the time of the campaign ranged from 2.8% among individuals not vaccinated at all to 10.7% among individuals with two doses, and for some, this illness was detected at the vaccination site (Table 5).

Fear of needles was more commonly cited as a reason for non-vaccination than a fear of side effects or the perception that the vaccine is dangerous (Table 5). Fears or concerns were more common among those not vaccinated at all (17.5%) than individuals vaccinated at least once (7.3%–7.8%, Table 5). Few individuals reported that getting sick with hepatitis E is not serious as a reason for non-vaccination.

**Table 5. Reasons for non-vaccination by number of doses received.**

| Reasons for non-vaccination | 0 doses | | 1 dose | | 2 doses | |
|---|---|---|---|---|---|---|
| **Physical absence** | **133** | **59.6%** | **176** | **80.7%** | **179** | **76.8%** |
| I was living in Bentiu IDP camp but was temporarily away | 62 | 27.8% | 109 | 50.0% | 129 | 55.4% |
| I was not living in Bentiu IDP camp during the vaccination | 46 | 20.6% | 60 | 27.5% | 27 | 11.6% |
| I was living in Bentiu IDP camp but not available during the hours | 25 | 11.2% | 7 | 3.2% | 23 | 9.9% |
| **Campaign awareness or organisation** | **9** | **4.0%** | **6** | **2.8%** | **2** | **0.9%** |
| I did not know about the vaccination campaign | 7 | 3.1% | 5 | 2.3% | 1 | 0.4% |
| I did not think that I was eligible for vaccination | 2 | 0.9% | 1 | 0.5% | 1 | 0.4% |
| I went to the vaccination site but there was no vaccine | 0 | 0.0% | 0 | 0.0% | | 0.0% |
| **Health issues** | **11** | **4.9%** | **6** | **2.8%** | **25** | **10.7%** |
| I was sick during the vaccination campaign | 8 | 3.6% | 5 | 2.3% | 21 | 9.0% |
| I went to the site but could not get vaccinated due to my health | 3 | 1.3% | 1 | 0.5% | 4 | 1.7% |
| **Fears and concerns** | **39** | **17.5%** | **17** | **7.8%** | **17** | **7.3%** |
| I am afraid of needles | 13 | 5.8% | 11 | 5.0% | 5 | 2.1% |
| I already received too many vaccines | 9 | 4.0% | 1 | 0.5% | 2 | 0.9% |
| I am worried about side effects | 4 | 1.8% | 4 | 1.8% | 7 | 3.0% |
| I think the vaccine is dangerous | 6 | 2.7% | 0 | 0.0% | 1 | 0.4% |
| I do not think getting sick with hepatitis E is serious | 6 | 2.7% | 0 | 0.0% | 1 | 0.4% |
| I am worried that the vaccine will affect my fertility in the future | 1 | 0.4% | 1 | 0.5% | 0 | 0.0% |
| My spouse / decision maker refused the vaccination | 0 | 0.0% | 0 | 0.0% | 1 | 0.4% |
| I have religious concerns | 0 | 0.0% | 0 | 0.0% | 0 | 0.0% |
| **Other** | | | | | | |
| Refused with no reason given | 32 | 14.3% | 13 | 6.0% | 11 | 4.7% |
| Unknown | 8 | 3.6% | 1 | 0.5% | 0 | 0.0% |
| Other | 4 | 1.8% | 2 | 0.9% | 2 | 0.9% |
| **Total n** | **223** | | **218** | | **233** | |

Non-mutually exclusive categories, individuals may have given more than one reason.

Among the 66 pregnant women who were not fully vaccinated, the distribution of reasons for non vaccination were similar to the non-pregnant population; most pregnant women reported that non-vaccination was related to absence (60%), while 9% missed a dose because of feeling sick due to pregnancy. Some did explicitly state fears related to side effects, fertility or pregnancy (6%), or fear of needles (3%), and one woman reported that she did not believe the vaccine was effective.

## Pharmacovigilance

**Passive AEFI surveillance.** Overall, the passive AEFI surveillance system detected 11 events after vaccination: two cases of fever after the first round of vaccination in March; none after round two in April and nine after the third round of vaccination in October.

Among survey participants reporting AEFI after the third round, 7 (78%) were female, 1 (11%) was male and 1 (11%) did not have sex documented. The age of these respondents ranged from 24 to 45 years, surpassing the upper limit of age eligibility for vaccination. The AEFI was reported after the third dose for 6 (67%) people, after the first or second dose for one person each and after unknown number of doses for one person. The events included 4 local reactions, 1 allergic reaction, 1 with fever, joint pain and vomiting, 2 others with details not specified and one pregnant woman with abdominal cramping. The pregnant woman was

**Table 6. Symptoms reported after dose of Hecolin vaccine in October 2022, n = 91 individuals reported symptoms after vaccination (each individual may have reported more than one symptom) among N = 1,195 individuals vaccinated.**

|  | n | % | Incidence per 100* |
|---|---|---|---|
| **Local reactions** | **17** |  | **1.42%** |
| Local swelling around injection site | 2 | 2.2% | 0.17% |
| Local pain around injection site | 11 | 12.1% | 0.92% |
| Local itch around injection site | 1 | 1.1% | 0.08% |
| Redness around injection site | 0 | 0.0% | 0.00% |
| Rash | 3 | 3.3% | 0.25% |
| **Systemic reactions** | **140** |  | **11.72%** |
| Fever | 68 | 74.7% | 5.69% |
| Headache | 29 | 31.9% | 2.43% |
| Fatigue and general weakness | 21 | 23.1% | 1.76% |
| Cough | 5 | 5.5% | 0.42% |
| Muscle aches | 9 | 9.9% | 0.75% |
| Nausea and/or vomiting | 2 | 2.2% | 0.17% |
| Diarrhea | 5 | 5.5% | 0.42% |
| **Other, specify** | **14** | **15.4%** | **1.17%** |

Non-mutually exclusive, individuals reported more than one symptom.

*calculated as n reporting symptom / total N vaccinated in third round.

observed at the MSF hospital and left in good condition after two hours. All individuals detected through this system were documented to have recovered.

**Survey results on new symptoms within 72 hours of vaccination.** Overall, 91 (7.6%) of 1,195 individuals who reported receiving a dose of the Hecolin vaccine in October 2022 (third round) reported a new symptom in the 72 hours after receiving the vaccine. Women more frequently reported experiencing new symptoms than men (73 (10%), vs. 18 (3.7%) p<0.001).

Local reactions were reported less frequently than systemic reactions, with 11 (12.1%) individuals reported local pain around injection site (0.92% incidence), 3 (3.3%) individuals reported rash (0.25% incidence), and 2 (2.2%) reported local swelling around injection site (0.17% incidence) (Table 6). Fever was the most common symptom reported by 68 (74.7% of individuals with symptoms, incidence of 5.69%), followed by headache (32%, incidence of 2.43%) and fatigue and general weakness (23%, incidence of 1.76%). In the "other" category, symptoms reported included "pain in the throat" (n = 1), "sleeping" (n = 4), "tested positive for malaria" (n = 4), "back pain" (n = 2), "heart burn" (n = 3), and "reported miscarriage next day" (n = 1). No hospitalisations or deaths were reported.

We conducted a sensitivity analysis including only individuals self-reporting symptoms after vaccination; we found an almost two-fold higher incidence of AEFI overall (14.3%, n = 58/407 compared to 7.6% n = 91/1,195, p<0.001) and no significant difference in reported symptoms by sex (11.8% among men vs. 14.9% among women, p = 0.461).

## Discussion

Our survey following the first mass reactive vaccination campaign in Bentiu IDP camp found that the hepatitis E vaccine was highly accepted. We estimate that 86% (95% CI 84–88%) of the vaccine-eligible population had at least one dose, 73% (95% CI 70–75) had two or more doses

and 58% (95% CI 55–61%) had the complete three dose schedule. There were no significant differences in vaccination coverage by age, sex, or camp sector of residence.

Most reasons for non-vaccination were related to absence rather than fear of needles or side effects reflecting a general positive attitude towards vaccination in the community. Although the campaign took place during the daytime, vaccination started early in the morning and continued the weekends, and unavailability during the hours of the vaccination campaign was less frequently the reason for non-vaccination. Population mobility is a challenge for accurately estimating population size and providing a multi-dose vaccination schedule over a six-month period. Among individuals who had one dose, around one third reported not residing in Bentiu IDP camp during the prior campaigns and half were temporarily away. Temporary absence was common accounting for over 50% of non-vaccination among individuals vaccinated at least once, and 55% among those with two doses. Many individuals were able to get only one dose during the last round in October because they returned to Bentiu IDP camp after the second round of vaccination in April. With limited employment opportunities and food availability in Bentiu IDP camp, residents frequently seek opportunities outside the camp, and many individuals maintain a second residence in their county of origin. This may partly explain the discrepancy in administrative and survey estimates of coverage, however, the similar estimates for the third dose (53% administrative coverage third dose vs. 58% survey estimate) likely reflect the real coverage among the population present at the time.

In our study, we captured few AEFI by passive surveillance at vaccination sites and health facilities, whereas the inclusion of questions about symptoms after vaccination revealed more symptoms. Overall, 7.6% of individuals vaccinated in October experienced at least one new symptom in the 72 hours after vaccination. Either individuals did not seek care for the symptoms reported in the survey, or we did not capture them in our surveillance system. Both care-seeking and reporting may be related to the severity of symptoms, and we hypothesize that more severe events would have been more likely captured by the surveillance and reported in the survey. We were unable to investigate the relatedness of the reported new symptom to the vaccine or categorize the type or severity of the symptoms in the survey and consider all as possible vaccine reactions. From the survey results, we estimated an incidence of local reactions of 1.4% and 11.7% systemic reactions. However, the AEFI assessment in our survey was a one-time interview conducted a mean of 23 days after vaccination and we accepted responses from a delegate. We found higher incidence of AEFI when looking only at individuals self-reporting their symptoms in the survey. Furthermore, we did not require a precise temperature definition for fever in the survey, and we did not attempt to compare this incidence to a non-vaccinated control group.

In the Hecolin phase 3 clinical trial, all participants were observed for 30 minutes after each vaccine dose [1]. A reactogenicity subset in both the vaccine and placebo groups were visited seven times at home at between 6 hours and 28 days after each dose. The remaining participants were asked to report any adverse events to a local clinic within one month of each vaccine dose. Both active and passive follow-up estimated a higher incidence of local adverse reactions in the Hecolin vaccine group compared to placebo (active follow-up: 13.5% Hecolin vs 7.5% placebo; passive follow-up: Hecolin 2.8% vs. 1.9% placebo) but a similar incidence of systemic adverse reactions in both groups (active follow-up: 20.3% Hecolin vs 19.8% placebo; passive follow-up: 1.9%). In our study, we found an incidence of systemic reactions (11.7%) higher than in the clinical trial passive follow-up, but lower than in the active follow-up. These systemic reactions (i.e. fever, headache, fatigue) could reflect the different measurement approach (i.e. ours based on a retrospective survey), or alternatively, a higher burden of other illnesses (e.g. malaria) in the Bentiu IDP camp population compared to the clinical trial population in China.

A major limitation of this survey is the high proportion of delegate respondents. This is a challenge in conducting a rapid survey after an intervention in a mobile adult working-age

population. When restricting the analysis to people self-reporting only, we found a similar overall vaccination coverage, but a higher incidence of AEFI. This higher incidence of AEFI however, remains comparable with what was reported in Hecolin clinical trials. We conducted the survey soon after the end of the third-round campaign, with interviews a median of 23 days (range 3–36 days) after respondents were vaccinated in the third round. There were multiple vaccination campaigns conducted at different times in Bentiu IDP camp in 2022, including for SARS-CoV-2, cholera and measles and it is possible that individuals reported vaccination for a different antigen. To mitigate this risk, interviewers showed a picture of the Hecolin vaccine, whose single dose presentation is unique, a picture of the vaccination card (a unique green colour with the MSF logo) and gave precise dates of the vaccination campaigns to ensure that responses referred to the hepatitis E vaccine specifically.

Vaccine confidence and acceptance is context specific and varies based on antigen and approach (i.e., reactive or routine vaccination), and should be explored using qualitative methods. The rapid deployment of new COVID-19 vaccines, the global scale of the COVID-19 pandemic and the proliferation of rumours through social media contributed to a context of heightened vaccine hesitancy in general during the campaign period [13]. Despite this general context, we found a high uptake of the Hecolin vaccine among the eligible population in Bentiu, which may reflect the almost decade-long transmission of hepatitis E in Bentiu IDP camp. Due to limited vaccine and license restrictions, this high coverage among the vaccine-eligible population corresponds to only 35% coverage with at least one-dose, and 24% complete dose schedule coverage in the full camp population. If the safety of the vaccine in children under the age of 16 years old were established, revision of license restrictions to include children under 16 years old could protect more of the population at risk in this setting.

## Conclusion

We found a high coverage of at least one dose of the Hecolin vaccine among the eligible population following three rounds of vaccination, and no severe adverse events following immunisation. We demonstrate that the vaccine was well accepted and well tolerated in the Bentiu IDP camp community after the first mass reactive vaccination campaign. To maintain high coverage over time, providing vaccines outside of mass reactive vaccination campaigns would allow the highly mobile population to be vaccinated according to their own timeline and help to prevent future outbreaks.

## Supporting information

**S1 Text. Supplementary tables. Table A.** Vaccination coverage with exact number of doses according to recall and by card, n = 1599. **Table B.** Sensitivity analysis including only those who self-reported, n = 513. **Table C.** Sensitivity analysis of vaccination coverage by dose according to recall and card, n = 1470 individuals 16–39 yrs.
(DOCX)

**S1 Fig.** Population pyramid for all individuals counted at randomly selected households (A) and vaccine-eligible individuals counted at randomly selected households (B).
(TIF)

## Acknowledgments

We thank the community of Bentiu IDP camp for their participation in the survey. We thank all the survey team members who conducted the interviews. We thank the Bentiu IDP camp

management and health partners, including the state ministry of health and primary health care centers for their support.

## Author Contributions

**Conceptualization:** Robin C. Nesbitt, Etienne Gignoux, Andrew S. Azman, Iza Ciglenecki.

**Data curation:** Robin C. Nesbitt, Vincent Kinya Asilaza.

**Formal analysis:** Robin C. Nesbitt.

**Methodology:** Robin C. Nesbitt, Etienne Gignoux, Aybüke Koyuncu, Andrew S. Azman.

**Project administration:** Robin C. Nesbitt, Priscillah Gitahi, Patrick Nkemenang, Jetske Duncker, Zelie Antier.

**Supervision:** Vincent Kinya Asilaza, Etienne Gignoux, Duol Biem, John Rumunu, Iza Ciglenecki.

**Writing – original draft:** Robin C. Nesbitt, Vincent Kinya Asilaza, Andrew S. Azman.

**Writing – review & editing:** Vincent Kinya Asilaza, Etienne Gignoux, Aybüke Koyuncu, Priscillah Gitahi, Patrick Nkemenang, Jetske Duncker, Zelie Antier, Melat Haile, Primitive Gakima, Joseph F. Wamala, Fredrick Beden Loro, Monica Rull, Andrew S. Azman, John Rumunu, Iza Ciglenecki.

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
