## [Decision Letter · Decision Letter 0]

26 Oct 2023

Dear Dr. Nesbitt,

Thank you very much for submitting your manuscript "Vaccination coverage and adverse events following a reactive vaccination campaign against hepatitis E in Bentiu displaced persons camp, South Sudan" for consideration at PLOS Neglected Tropical Diseases. As with all papers reviewed by the journal, your manuscript was reviewed by members of the editorial board and by several independent reviewers. In light of the reviews (below this email), we would like to invite the resubmission of a significantly-revised version that takes into account the reviewers' comments. 

This is an interesting article on the vaccination coverage and adverse events following a mass vaccination campaign.

I would be grateful if the authors could respond to the comments from reviewers including from reviewer 3, would it be possible to provide data on efficacy?

We cannot make any decision about publication until we have seen the revised manuscript and your response to the reviewers' comments. Your revised manuscript is also likely to be sent to reviewers for further evaluation.

Sincerely,

Alexander Stockdale, PhD MRCP

Academic Editor

Michael Holbrook

Section Editor

Reviewer's Responses to Questions

Reviewer #1: In the manuscript entitled “Vaccination coverage and adverse events following a reactive vaccination campaign against hepatitis E in Bentiu displaced persons camp, South Sudan”, Robin Nesbitt et al. reported the vaccination coverage and occurrence of AEFI through a sampling survey conducted in the context of mass hepatitis E vaccination campaigns in South Sudan. First of all, it must be acknowledged that the reactive vaccination campaign against hepatitis E in Bentiu internally displaced camp is an immensely commendable and challenging job. It not only would effectively reduce the incidence of hepatitis E in this high-prevalence area but also diminish the burden of severe illness and death caused by hepatitis E. Furthermore, it sets a precedent for future reactive vaccination campaigns in other regions. The author's investigation of vaccination coverage and AEFI conducted after this vaccination campaign is also of paramount importance. Firstly, having a relatively precise understanding of vaccination coverage provides a basis for exploring vaccine effectiveness in the future. Secondly, hepatitis E vaccine had not been widely used in high-prevalence areas or among pregnant women before this campaign, making this safety data particularly significant. Considering the challenges of conducting surveys in this highly mobile population, at this stage, my comments on this manuscript are as follows:

Major Comments

1. Methods: 

a. Line 95-100: It would be helpful if the authors could provide additional details regarding the sample size calculation, ideally including the formula used for the calculation. Additionally, explaining the meaning of design effect and how "deff=2" was determined, possibly with references, would help readers better understand the results of this study. And I haven’t found any explanation from the authors in either the results or discussion sections regarding the meaning of "Deff" in Tables 3 and 4.

b. Could the authors please clarify the definitions of “the target population” and “the vaccine-eligible population”, and the relationship between them in the Methods section? The sentence at line 100 “The target population for the survey was vaccine-eligible individuals” makes it confusing to read lines 162-166 of the Results. Which population's coverage is reported as the survey's result, and which is estimated based on the survey's findings?

2. Results

a. Line 157-160: The authors presented survey results on “previous hepatitis E infection” in the results section. What is the purpose and significance of this survey? I didn’t find any relevant explanations in the Methods or Discussion.

b. Line 219-225: any AE should be expressed as an incidence rate rather than a proportion among those who had at least one AE.

c. Table 6: the 'n' for systemic reactions is 140, which exceeds the 91 people reported for any symptom? 

d. The authors should double-check all the data in this manuscript to ensure the presented data are correct. The multiple numbers in the text are quite confusing, for example, the several instances of the term "target", such as lines 86-88, line100, line 149, line 295 and those in Table 1, should be clarified by the authors that if these “target” refer to the same population (it seems not while the numbers before and after do not align consistently). If not, the authors should consider rewording and adding footnote of the tables to provide greater clarity. 

3. Discussion

a. I recommend that the authors further expand on the limitations of this study, including but not limited to the retrospective survey method that may lead to recall bias, the lack of a precise temperature definition for fever in the survey results, and no control group who didn’t vaccinated with hepatitis E vaccine to compare the incidence of AE.

b. Line 284-286: The authors mentioned that “There were multiple vaccination campaigns conducted in Bentiu IDP camp in 2022, including for SARS-CoV-2, cholera and measles and it is possible that individuals reported vaccination for a different antigen.”. This could not only potentially lead to vaccinees confusing the source of adverse reactions but also the concurrent use of different vaccines in a short time frame (even on the same day, if possible?) might increase adverse reactions to vaccines. The authors should clarify if such a possibility exists.

Minor Comments

1. There are several abbreviations in the main text without fully explanations. Please review the entire text and make the necessary revisions.

2. Line 61: the vaccine efficacy of three doses of Hecolin over the first 4.5 years was 93.3% (95%CI 78.6–97.9). If the authors intend to present the VE of a single dose of hepatitis E vaccine, it is important to clearly specify "single-dose efficacy" in this sentence.

3. Line 112-114: repeated contents.

4. Line 147: how to understand “abandoned”?

5. Line 214: this subheading should be improved to be clearer and more understandable.

Reviewer #2: Use of this vaccine in an outbreak is long overdue and hopefully this will give confidence to other decision-makers to deploy this vaccine early in an outbreak and prevent unnecessary morbidity and mortality, especially among pregnant women. Other than the error on line 112, I believe this manuscript to be fully ready for publication.

Reviewer 3:

Nesbitt et al in their manuscript titled “Vaccination coverage and adverse events following a reactive vaccination campaign against hepatitis E in Bentiu displaced persons camp, South Sudan” conducted a vaccination coverage survey on Hepatitis E using simple random sampling from the camp shelters following the third round of vaccination. They conducted the passive surveillance and reported the adverse events following immunisation (AEFI) on the reactive vaccine for peoples displaced in Bentiu IDP camp, South Sudan, and concluded that no severe AEFI were observed in people administered with hepatitis E vaccine of one to three doses in the Bentiu IDP camp community and suggested that the vaccine (Helicon) was accepted and well tolerated and should be considered for use in future outbreak response. Although the surveillance is on the relatively large-scale population with a sound conclusion and an useful information, I don’t believe that the contents is complete in term of reactive vaccination of Helicon in the population since the manuscript is only concerned with the safety. From this aspect, the manuscript should also include the efficacy aspect and therefore it is far from reaching the standard for this journal. 

Minor correction

Line 20: Hepatitis E (HEV) genotypes 1 and 2 are a common… change to “Hepatitis E (HEV) genotypes 1 and 2 are the common…”

Line 233: …… found that the hepatitis E vaccine was highly accepted, we estimate that 86% (95% CI 84-88%) of the vaccine-eligible ….. change to “…… found that the hepatitis E vaccine was highly accepted. We estimate that 86% (95% CI 84-88%) of the vaccine-eligible …..”

PLOS authors have the option to publish the peer review history of their article (what does this mean?). If published, this will include your full peer review and any attached files.

Reviewer #1: No

Reviewer #2: No

Reviewer #3: No

**Key Review Criteria Required for Acceptance?**

**Methods**

-Are the objectives of the study clearly articulated with a clear testable hypothesis stated?

-Is the study design appropriate to address the stated objectives?

-Is the population clearly described and appropriate for the hypothesis being tested?

-Is the sample size sufficient to ensure adequate power to address the hypothesis being tested?

-Were correct statistical analysis used to support conclusions?

-Are there concerns about ethical or regulatory requirements being met?

Reviewer #2: The objectives and methods are clear in this descriptive analysis. The population is clearly described and correct sample size and statistical analysis were used. No ethical concerns.

**Results**

-Does the analysis presented match the analysis plan?

-Are the results clearly and completely presented?

-Are the figures (Tables, Images) of sufficient quality for clarity?

Reviewer #2: The analysis presented matched the plan and the results are clearly and simply presented. Tables and figures are clear and appropriate.

**Conclusions**

-Are the conclusions supported by the data presented?

-Are the limitations of analysis clearly described?

-Do the authors discuss how these data can be helpful to advance our understanding of the topic under study?

-Is public health relevance addressed?

Reviewer #2: The conclusions are supported by the data. The limitations are discussed. This is important data to show the acceptability and feasibility (though with limited multi-dose coverage) of deploying Hecolin into an outbreak setting in an IDP camp.

**Editorial and Data Presentation Modifications?**

Reviewer #2: there is an error on line 112 where a statement is inadvertently repeated. Likely just missed in editing.
---

## [Editor Report · Decision Letter 1]

8 Jan 2024

Dear Dr. Nesbitt,

We are pleased to inform you that your manuscript 'Vaccination coverage and adverse events following a reactive vaccination campaign against hepatitis E in Bentiu displaced persons camp, South Sudan' has been provisionally accepted for publication in PLOS Neglected Tropical Diseases.

Best regards,

Alexander Stockdale, PhD MRCP

Academic Editor

Michael Holbrook

Section Editor

---

## [Editor Report · Acceptance letter]

18 Jan 2024

Dear Dr. Nesbitt,

We are delighted to inform you that your manuscript, "Vaccination coverage and adverse events following a reactive vaccination campaign against hepatitis E in Bentiu displaced persons camp, South Sudan," has been formally accepted for publication in PLOS Neglected Tropical Diseases.

Best regards,

Shaden Kamhawi

co-Editor-in-Chief

Paul Brindley

co-Editor-in-Chief
